# TAROT: Targeted Data Selection via Optimal Transport

Lan Feng [1] [*]   Fan Nie [2] [*]   Yuejiang Liu [2]   Alexandre Alahi [1]

## Abstract

We propose TAROT, a **Tar**geted data selection framework grounded in **O**ptimal **T**ransport theory. Previous targeted data selection methods primarily use influence-based greedy heuristics to enhance domain-specific performance. These methods perform well on limited, unimodal data (i.e., data following a single pattern) but become less effective as target data increases in complexity. Specifically, in multimodal distributions, these heuristics fail to account for multiple inherent patterns, leading to suboptimal data selection. This work identifies two primary factors contributing to this limitation: (i) the disproportionate impact of dominant feature components in high-dimensional influence estimation, and (ii) the restrictive linear additive assumptions inherent in greedy selection strategies. To address these challenges, TAROT incorporates whitened feature distance to mitigate dominant feature bias, offering a more reliable measure of data influence. Building on this, TAROT uses whitened feature distance to quantify and minimize the optimal transport distance between the selected data and target domains. Notably, this minimization also facilitates the estimation of optimal selection ratios. We evaluate TAROT across multiple tasks, including semantic segmentation, motion prediction, and instruction tuning. Results consistently show that TAROT outperforms state-of-the-art methods, highlighting its versatility across various deep learning tasks. Code is available at: https://github.com/vita-epfl/TAROT.

## 1. Introduction

Expanding model capacity and data volume has emerged as a key strategy in both computer vision (CV) and large

*Equal contribution [1]EPFL, Switzerlanzd [2]Stanford, USA. Correspondence to: Yuejiang Liu <yuejiang.liu@stanford.edu>, Alexandre Alahi <alexandre.alahi@epfl.ch>.

*Proceedings of the 42nd International Conference on Machine Learning*, Vancouver, Canada. PMLR 267, 2025. Copyright 2025 by the author(s).

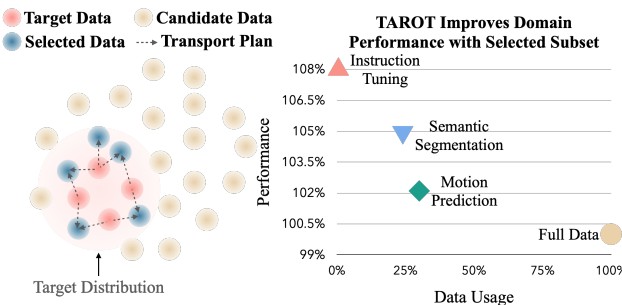

Figure 1: **Left:** the selection objective of TAROT, designed to minimize optimal transport costs. The method operates on a target distribution and a candidate data pool, selecting a subset from the candidate data to reduce transport costs. **Right:** the approximate performance improvements and data efficiency achieved by TAROT across various tasks. It illustrates that TAROT consistently enhances task performance in different domains while requiring less data. Detailed experimental results are provided in Section 4.

language models (LLMs) (Achiam et al., 2023; Zou et al., 2024; Oquab et al., 2023; Feng et al., 2024). However, training a model on all available data may not be optimal or feasible, as data quality varies and computational resources are often limited (Albalak et al., 2024; Liu et al., 2024; Engstrom et al., 2024; Kang et al., 2024). To address this, recent research has introduced targeted data selection methods. These methods aim to select data from a candidate pool that maximizes model performance on specific target tasks. Recent methods (Xia et al.; Engstrom et al., 2024) address this challenge using a two-step approach: (1) **influence estimation**, which calculates data influence scores between the candidate and target datasets, and (2) **greedy heuristics**, which selects data with the highest average influence scores for the target dataset.

Although these selection paradigms have shown to be effective in LLM applications (Engstrom et al., 2024; Xia et al.), they rely heavily on the assumption that influence functions are additive. This means that the influence of multiple training samples on a target sample can be added together. However, as noted by Hu et al. (2024), this assumption, while efficient and broadly applicable, fails even in simple linear regression scenarios. One main reason why the additivity assumption works well in some cases is the simple and unimodal distribution of the target domain, such

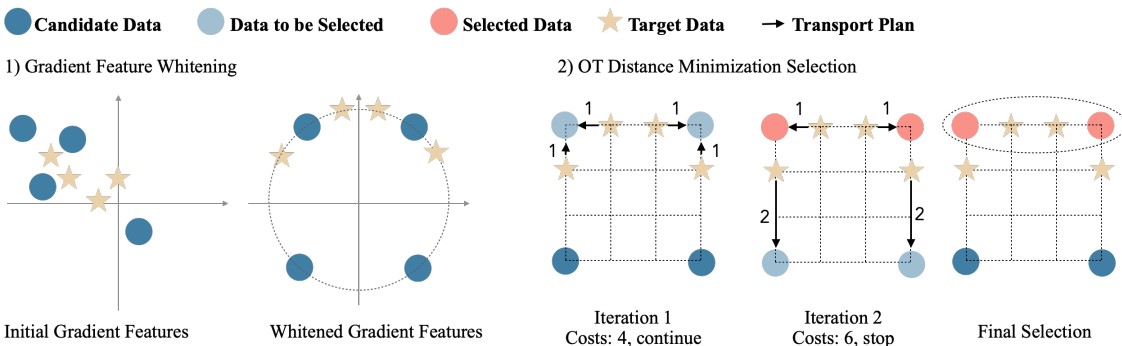

Figure 2: **Overview of TAROT. Left:** Raw gradient features are whitened and projected onto a unit hypersphere. **Right:** A greedy algorithm iteratively selects candidate samples. During the second iteration, adding two new distant data points increases the overall transport cost, prompting the algorithm to stop. As a result, only in-distribution data points are selected to minimize the OT distance.

as a single task (e.g. code generation) in LLMs. In these situations, adding up the influences provides a good estimate of how a candidate sample impacts the entire target task. However, in other contexts like motion prediction, where data distributions are often complex and multimodal, this approach falls short. In such cases, the summed influence does not capture the diverse nature of the target distribution, reducing its effectiveness.

To this end, we adopt a distribution-matching approach for targeted data selection. We first introduce *whitened feature distance* (**WFD**), an influence estimation method designed to mitigate the bias from dominant features. The benefits of feature whitening are also discussed in (Ermolov et al., 2021). Using **WFD** as pairwise cost estimation between data points, we develop TAROT, an Optimal Transport (OT)-based targeted selection framework. This method selects data that minimizes the OT distance between the target distribution and the selected samples. Figure 1 illustrates the selection objective of our approach. Our choice of OT is also supported by findings in (Just et al., 2023), which demonstrate that training models on source data with smaller OT distances to the target domain improves domain-specific performance.

To validate the general effectiveness of our method, we conduct experiments on semantic segmentation, motion prediction, and instruction tuning tasks. Extensive results demonstrate that our approach consistently outperforms previous methods. Notably, models trained on smaller, domain-relevant datasets often surpass those trained on larger, less relevant datasets. In summary, our contributions are:

- **Improved influence estimation**: we present **WFD** as a reliable measure of data influence.
- **Distribution matching-based data selection**: we propose TAROT, an OT-based pipeline for targeted data selection that reduces distributional differences between the target and selected datasets.

- **Broad applicability and improved performance**: extensive experiments in CV and LLMs demonstrate that TAROT consistently outperforms existing methods, allowing models trained on smaller, targeted datasets to match or surpass the performance of those trained on full datasets.

## 2. Preliminary

### 2.1. Data Influence Formulation

Following Pruthi et al. (2020), we summarize how to estimate the influence of a training data point on held-out data using a first-order approximation of training dynamics.

**First-order Approximation to Per-step Influence.** Suppose $x_i$ is the only training sample at time step $t$, and the model $\theta^t$ is updated based on the loss $\mathcal{L}(x_i; \theta^t)$ using SGD with learning rate $\eta_t$. The per-step influence of $x_i$ on a target sample $x$ is defined as the reduction in the loss of $x$ caused by training on $x_i$. This can be approximated using a first-order Taylor expansion. The per-step influence is:

$$-\eta_t \langle \nabla \mathcal{L}(x_i; \theta^t), \nabla \mathcal{L}(x; \theta^t) \rangle, \qquad (1)$$

where $\nabla \mathcal{L}(x_i, x; \theta^t)$ is the loss gradient.

**Trajectory Influence via Checkpoints.** For practicality, Pruthi et al. (2020) compute the trajectory influence by summing the per-step influences over epochs:

$$\text{Inf}_{\text{SGD}}(x_i, x) \triangleq \sum_{i=1}^{N} \overline{\eta}_i \langle \nabla \mathcal{L}(x; \theta_i), \nabla \mathcal{L}(x_i; \theta_i) \rangle. \quad (2)$$

### 2.2. Background on Optimal Transport

Given a complete, separable metric space $\mathcal{X}$ and probability measures $\alpha, \beta \in \mathcal{P}(\mathcal{X})$, the Kantorovich formulation (Kantorovitch, 1958) of OT is defined as:

$$\text{OT}(\alpha, \beta) \triangleq \min_{\pi \in \Pi(\alpha, \beta)} \int_{\mathcal{X} \times \mathcal{X}} c(x, y) \, d\pi(x, y), \quad (3)$$

where $c(\cdot, \cdot) : \mathcal{X} \times \mathcal{X} \to \mathbb{R}^+$ is a cost function, and $\Pi(\alpha, \beta)$ is the set of couplings with marginals $\alpha$ and $\beta$.

In practice, the measures $\alpha$ and $\beta$ are approximated using finite samples $\{\mathbf{x}^{(i)}\}$ and $\{\mathbf{y}^{(j)}\}$, with corresponding discrete measures $\alpha = \sum_{i=1}^{n} a_i \delta_{\mathbf{x}^{(i)}}$ and $\beta = \sum_{j=1}^{m} b_j \delta_{\mathbf{y}^{(j)}}$. The pairwise costs form an $n \times m$ matrix $\mathbf{C}$, where $\mathbf{C}_{ij} = c(\mathbf{x}^{(i)}, \mathbf{y}^{(j)})$. Solving this problem as a linear program is computationally expensive, with cubic complexity. In implementation, we use the Sinkhorn algorithm (Cuturi, 2013) for efficient calculation.

## 3. Method

The failure modes of *influence-based greedy heuristics* have been systematically studied in (Hu et al., 2024). Specifically, errors in the influence function and the inability to account for the non-additive nature of collective influence can cause these heuristics to fail even in simple linear regression tasks.

In contrast, we formulate targeted data selection as a distribution matching problem, aiming to select a subset of candidate data that closely aligns with the target distribution. We employ the OT distance to quantify the discrepancy between distributions.

Sections 3.1 and 3.2 demonstrate how to measure the OT distance between datasets, and Section 3.3 discusses strategies for selecting data to minimize the OT distance. An overview of these processes is illustrated in Figure 2.

### 3.1. Optimal Transport Between Datasets

As discussed in (Alvarez-Melis & Fusi, 2020), estimating OT distance between datasets begins with defining a distance metric between data pairs $(x, y)$ and $(x', y')$. In Section 2.1, we introduced a gradient-based method for estimating data influence, which embeds both input features and labels into the model's gradient. This gradient feature captures the task-specific relationship between data points and the model's loss. By computing distances such as the L2 distance, it provides a viable metric for the OT problem. Furthermore, it aligns with the model's optimization objective, ensuring that the OT distance reflects meaningful differences between datasets in terms of their impact on model training.

Formally, we define a dataset $\mathcal{D}$ as a set of feature-label pairs $(x, y) \in \mathcal{X} \times \mathcal{Y}$, where $\mathcal{X}$ is the feature space and $\mathcal{Y}$ is the label set. For convenience, we use the shorthand $z \triangleq (x, y)$ and $\mathcal{Z} \triangleq \mathcal{X} \times \mathcal{Y}$. We consider a candidate dataset $\mathcal{D}_c = \left\{ z_c^{(i)} \right\}_{i=1}^{N}$ and a target dataset $\mathcal{D}_t = \left\{ z_t^{(i)} \right\}_{i=1}^{M}$.

We embed $z$ using the model's parameters trained on $\mathcal{D}_c$:

$$\phi(z) = \sum_{i=1}^{T} \nabla \mathcal{L}(z; \theta_i), \tag{4}$$

where $T$ is the total number of checkpoints, and $\nabla \mathcal{L}(z; \theta_i)$ represents the gradient of the loss function with respect to the model parameters at the $i$-th checkpoint $\theta_i$.

The distance between two samples is then defined as:

$$d_{\mathcal{Z}}(z, z') = \|\phi(z) - \phi(z')\|_2. \tag{5}$$

### 3.2. Whitened Feature Distance

Directly computing the distance $d_{\mathcal{Z}}(z, z')$ as defined in Equation (5) presents significant challenges due to the correlation of gradient features. This correlation results in an ill-conditioned covariance matrix, allowing certain feature space directions to dominate distance estimation and distort sample relationships. Additionally, varying gradient component scales bias distance measurements, impairing the accuracy of distance estimation.

To address these issues, we propose a novel application of *whitening* to the gradient features. Whitening effectively decorrelates the features and scales them to have unit variance, ensuring that each feature contributes equally to the distance computation. Our method comprises three steps:

1. **Random projection.** We adopt the same random projection technique from (Xia et al.; Park et al., 2023) to reduce the dimensionality of the gradients. The projected gradient $\phi(z)_i^{\text{proj}}$ is computed as:

$$\phi(z)_i^{\text{proj}} = \mathcal{P}^T \cdot \phi(z)_i, \tag{6}$$

where $\mathcal{P}^T$ is the projection matrix. We leverage the efficient CUDA implementation from Park et al. (2023).

2. **Whitening.** Despite dimensionality reduction, projected gradients remain correlated, creating an anisotropic feature space. To address this, we perform Cholesky whitening to decorrelate features. First, we center the projected gradients $\Phi(z)^{\text{proj}} = [\phi(z)_1^{\text{proj}}, \ldots, \phi(z)_N^{\text{proj}}]$:

$$\tilde{\phi}(z)_i^{\text{proj}} = \phi(z)_i^{\text{proj}} - \frac{1}{N} \sum_{i=1}^{N} \phi(z)_i^{\text{proj}}. \tag{7}$$

Next, we compute the covariance matrix $\Sigma$ and apply Cholesky decomposition, $\Sigma = LL^\top$, where $L$ is lower triangular. The gradients are whitened as:

$$\phi(z)^w = L^{-1} \tilde{\phi}(z)^{\text{proj}}, \tag{8}$$

ensuring decorrelation.

3. **Normalization.** Each whitened gradient vector $\phi(z)_i^w$ is normalized to unit length to ensure consistent gradient scales:

$$\hat{\phi}(z)_i = \frac{\phi(z)_i^w}{|\phi(z)_i^w|}. \tag{9}$$

The **WFD** $d_{\mathcal{Z}}^w(z, z')$ is then calculated as:

$$d_{\mathcal{Z}}^w(z, z') = \|\hat{\phi}(z) - \hat{\phi}(z')\|_2. \tag{10}$$

This gradient-based distance also applies to data influence estimation. Our experiments show that **WFD** outperforms the state-of-the-art method (Park et al., 2023), as detailed in Section 4.1.

Using $d_{\mathcal{Z}}^w$, we lift the metric to a dataset-level OT distance:

$$d_{\text{OT}}(\mathcal{D}_c, \mathcal{D}_t) = \min_{\pi \in \Pi(\alpha, \beta)} \int_{\mathcal{Z} \times \mathcal{Z}} d_{\mathcal{Z}}^w(z, z') \pi(z, z'). \tag{11}$$

### 3.3. Data Selection via OT Distance Minimization

Our objective is to select a subset $\mathcal{D}_s \subseteq \mathcal{D}_c$ of size $S \leq N$, such that the OT distance between the selected subset $\mathcal{D}_s$ and $\mathcal{D}_t$ is minimized.

Formally, let $\mathcal{D}_s = \left\{ z_s^{(i)} \right\}_{i=1}^{S}$ be a subset of $\mathcal{D}_c$. We aim to find $\mathcal{D}_s$ such that:

$$d_{\mathrm{OT}}(\mathcal{D}_s, \mathcal{D}_t) < d_{\mathrm{OT}}(\mathcal{D}_c, \mathcal{D}_t), \tag{12}$$

where $d_{\mathrm{OT}}$ is subject to the constraint that the masses for each point are normalized.

Solving the subset selection problem to minimize the OT distance is inherently combinatorial, as the number of possible subsets $\mathcal{D}_s \subseteq \mathcal{D}_c$ grows exponentially with the size of $\mathcal{D}_c$. An exhaustive search incurs prohibitive computational costs. To address this, we propose a greedy algorithm that efficiently approximates the solution by iteratively selecting elements that locally minimize the OT distance.

Our method exploits the sparsity of gradient-based features in high-dimensional space, which results in a sparse coupling matrix in the OT problem. Specifically, most of the transport mass in the OT plan is concentrated on the nearest neighbors of target points in $\mathcal{D}_t$. Leveraging this property, the algorithm iteratively selects the nearest neighbors from $\mathcal{D}_c$ for each point in $\mathcal{D}_t$, providing a computationally efficient solution.

Based on this, we introduce two selection schemes tailored for different scenarios: (1) fixed-size selection for maximizing performance under a limited budget, and (2) OT distance minimization selection to find a potentially optimal selection ratio.

**Fixed-Size Selection.** This scheme selects a fixed number of points $S$ from $\mathcal{D}_c$. For $M$ target samples, we iteratively add their nearest candidates from $\mathcal{D}_c$ to the selected set $\mathcal{D}_s$. In iteration $k$, the $k$-th nearest samples are considered. If adding all of them keeps $|\mathcal{D}_s| \leq S$, they are directly added, and the process continues. Otherwise, we compute the potential $\phi_i$ of the new candidates using the dual OT problem:

$$\min_{\pi \in \Pi(\mathcal{D}_s \cup \{z_i\}, \mathcal{D}_t)} \int_{\mathcal{D}_s \cup \{z_i\} \times \mathcal{D}_t} c(z, z_t) \, d\pi(z, z_t), \tag{13}$$

where $\Pi(\mathcal{D}_s \cup \{z_i\}, \mathcal{D}_t)$ is the set of couplings between $\mathcal{D}_s \cup \{z_i\}$ and $\mathcal{D}_t$. As noted in (Just et al., 2023), $\phi_i$ estimates the benefit of adding $z_i$ for minimizing the OT distance. We rank candidates by potential and select the top $S - |\mathcal{D}_s|$ samples. The process ends once these samples are added.

**OT-Distance Minimization Selection (OTM).** This scheme minimizes the OT distance between the selected data $\mathcal{D}_s$ and the target distribution. To avoid overfitting to specific points in $\mathcal{D}_t$, we use a $k$-fold selection strategy with cross-validation. The target dataset $\mathcal{D}_t$ is randomly split into $k$ equal subsets. In each fold, $1/k$ of $\mathcal{D}_t$ is used for selection,

while the OT distance is evaluated against the remaining $(k-1)/k$ data.

At each iteration, points from $\mathcal{D}_c$ are selected based on their nearest-neighbor distances to $1/k$ of $\mathcal{D}_t$, and the OT distance is monitored. The process stops when the OT distance increases:

$$d_{\mathrm{OT}}(\mathcal{D}_s, \mathcal{D}_t \setminus \mathcal{D}_t^{(fold)}) > d_{\mathrm{OT}}\left( \mathcal{D}_s^{(prev)}, \mathcal{D}_t \setminus \mathcal{D}_t^{(fold)} \right), \tag{14}$$

where $\mathcal{D}_t^{(fold)}$ is the $1/k$ subset used for selection.

After processing all $k$ folds, the selected subsets are combined to form the final dataset $\mathcal{D}_s$. The choice of $k$ balances distribution alignment and generalization. In our experiments, $k = 10$ ensures a good match with the target distribution while avoiding overfitting.

### 3.4. Data Weighting with OT Potential

To highlight the importance of individual data points, we assign weights based on their OT potentials from Equation (13). These potentials are scaled to positive integer weights, determining the number of times each point is duplicated within an epoch. The scaled weights $\{w_i\}$ are normalized to satisfy $\sum_{i=1}^{S} w_i = R$, where $R$ is a customizable repetition factor. For instance, $R$ can be set to $N + M$ to match the same training steps as using the full dataset, or adjusted based on specific training requirements, e.g., limited training budget.

## 4. Experiments

This section starts by evaluating the influence estimation performance of **WFD** in Section 4.1. Following that, we evaluate TAROT across diverse tasks, including semantic segmentation (Section 4.2), motion prediction (Section 4.3), and instruction tuning (Section 4.4).

### 4.1. WFD for Influence Estimation

**Evaluation Metric.** Following Park et al. (2023), we use the Linear Data Modeling Score (LDS) as our evaluation metric. LDS measures the effectiveness of an attribution or influence estimation method in predicting changes in model outputs when the training set is altered. Further details of LDS are provided in the supplementary materials.

**Target and Candidate Datasets.** Following Park et al. (2023), we evaluate image classification using ResNet-9 classifiers trained on the CIFAR-10 dataset. For motion prediction, we adopt AutoBots (Girgis et al., 2021), training on the nuScenes (Caesar et al., 2020) dataset (32k samples) and validating on 9k target samples.

**Baselines.** We compare **WFD** against the state-of-the-art data attribution method TRAK (Park et al., 2023) using the same ensemble approach. Since **WFD** quantifies distance (where smaller values indicate greater influence), we evaluate it with the LDS score by considering negative **WFD**

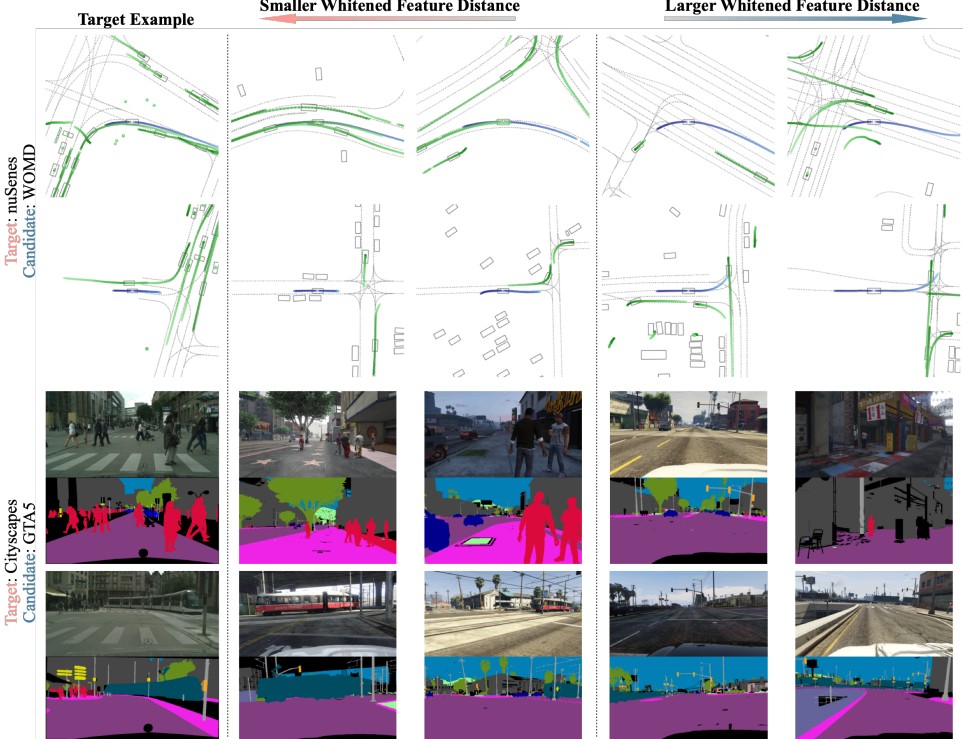

Figure 3: Illustration of target examples and their corresponding closest (most helpful) and furthest (most detracting) candidate examples, identified by **WFD**. **Top two rows**: Visualization of motion prediction tasks using the nuScenes and Waymo datasets. The dark-to-light blue lines indicate the past (input) and future (ground truth) trajectories for the target vehicle, while green lines represent surrounding vehicle trajectories. **Bottom two rows**: Semantic segmentation tasks with examples from the Cityscapes and GTA5 datasets. For each target example, the top image shows the RGB scene, and the bottom image depicts the corresponding ground truth semantic segmentation mask. The **WFD** method selects training examples that are semantically aligned with the target, demonstrating its effectiveness in identifying positive and negative samples across different tasks.

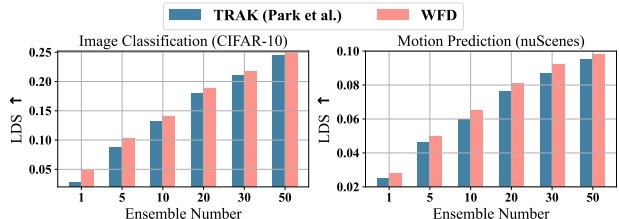

Figure 4: **WFD** outperforms baseline methods on both tasks. The x-axis indicates the number of model checkpoints that a given method uses to compute attribution scores. The y-axis indicates the method's efficacy as measured by LDS.

values. Additionally, we employ ZCA whitening instead of Cholesky whitening to preserve data structure for improved LDS performance. Differences between these two whitening methods are discussed in the supplementary materials.

**Main Results.** Figure 4 compares the performance of **WFD** with TRAK on the LDS score. Key findings include:

- **WFD outperforms TRAK regardless of ensemble sizes**.

This indicates that **WFD** exhibits a higher correlation with datamodels, which serves as an "oracle" of sorts for the LDS objective.

- **Inspecting WFD-identified examples.** In Figure 3 we display, for four randomly selected target samples from the nuScenes (Caesar et al., 2020) and Cityscapes (Cordts et al., 2016) dataset, the training examples from WOMD (Ettinger et al., 2021) and GTA5 (Richter et al., 2016) dataset corresponding to the smallest (most positive) and largest (most negative) whitened feature distance.

### 4.2. TAROT for Semantic Segmentation

Semantic segmentation involves assigning semantic labels to every pixel in an image. Prior studies show that synthetic data can significantly boost segmentation model performance on real-world data (Richter et al., 2016). We then address the following research question: **Can TAROT identify synthetic data that better align with real-world distributions to enhance model performance?**

**Datasets.** The GTA5 dataset (Richter et al., 2016) serves

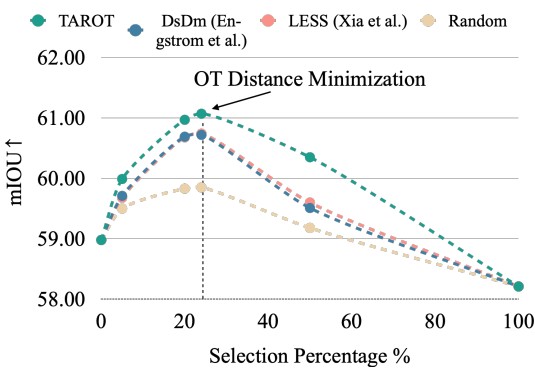

Figure 5: **Results of Targeted Selection for Semantic Segmentation.** Performance curves of DeepLabV3 trained using different data selection methods.

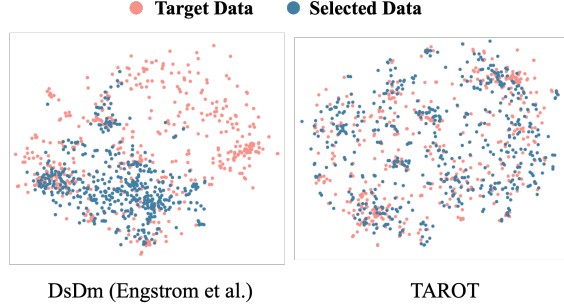

Figure 6: **Data selection visualization in gradient feature space with T-SNE.** We select 10% of Cityscapes' training set, with validation split as the target. **Left:** Samples selected by DsDm cluster tightly around the target distribution center. **Right:** Samples selected by TAROT are distributed more broadly, capturing the target distribution's complexity.

as the candidate dataset, while the Cityscapes (Cordts et al., 2016) training split (2975 samples) is used as the target dataset, with its validation split for evaluation. GTA5 contains 24,966 synthetic images with pixel-level annotations of urban scenes, whereas Cityscapes provides finely annotated real-world urban images.

**Baselines.** We compare TAROT with state-of-the-art targeted selection methods LESS and DsDm (Xia et al.; Engstrom et al., 2024), as well as random selection.

**Data Selection and Model Training.** We employ DeepLabV3 (Chen, 2017) with a ResNet50 (He et al., 2016) backbone, using gradients from the last four checkpoints to guide data selection. We evaluate selection ratios of 5%, 20%, 50% and OTM (Section 3.3). OTM selects approximately 24% of the data. The repetition factor is set as: $R = N + M$ for data selected by TAROT (Section 3.4), while other methods employ uniform repetition to match the same size. A new DeepLabV3 model is then trained from scratch on the selected subsets.

**Evaluation Metric.** Mean Intersection over Union (mIoU) is used as the primary evaluation metric. We report the best validation performance during training.

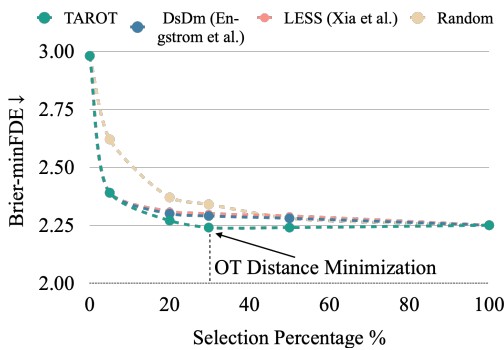

Figure 7: **Results of Targeted Selection for Motion Prediction.** The figure illustrates the performance of Wayformer under various data selection methods.

**Main Results.** Figure 5 compares the performance of TAROT and other baselines. Key findings include:

• **Some synthetic data can degrade model performance.** Training model on the full synthetic GTA5 dataset lowers mIoU from 58.98 to 58.21, highlighting the negative impact of misaligned synthetic data.

• **Targeted data selection enhances domain-specific performance.** Model performance improves with increased selection ratios but declines beyond an optimal point. Notably, TAROT consistently outperforms baselines, achieving the highest mIoU with about 24% of the GTA5 dataset. This demonstrates the effectiveness of OT distance in guiding optimal selection ratios.

• **Additional insights from T-SNE visualizations.** Figure 6 shows T-SNE visualizations of gradient features for selected data. The visualizations reveal differing selection behaviors: DsDm tightly clusters around the target distribution center, failing to capture its complexity. In contrast, TAROT selects data that better aligns with the broader target distribution, highlighting a key limitation of averaging-based influence methods in handling complex distributions.

### 4.3. TAROT for Motion Prediction

Motion prediction, the task of forecasting future trajectories from past data, is a critical component of autonomous driving. This section addresses two key research questions:

• **Is TAROT generic?** Compared to traditional computer vision tasks (e.g., image classification), motion prediction involves more complex inputs (e.g., maps, trajectories) and outputs (e.g., predicted trajectories and confidence scores). Is TAROT still effective in a more challenging setting?

• **Is TAROT transferable?** For example, can a model trained on data selected by a smaller model improve the performance of a larger model?

**Datasets.** We use the UniTraj framework (Feng et al., 2024) for unified training and evaluation across multiple datasets, including Waymo Open Motion (WOMD) (Ettinger et al.,

Table 1: Performance of TAROT on instruction tuning for LLMs. 'Transfer' indicates that QWEN-2.5-7B is trained on data selected by LLAMA-3.1-8B. Otherwise, the same base model is used for both selection and fine-tuning. 'All' denotes employing the full dataset. 5% indicates a selection ratio of 5%. The percentages within parentheses represent the selection ratios determined by OT-Minimization. 'Subtask label' specifies whether subtask labels are required during data selection.

| Dataset | LLAMA-3.1-8B | | QWEN-2.5-7B | | Transfer | | Subtask label |
| | MMLU $\uparrow$ | BBH $\uparrow$ | MMLU $\uparrow$ | BBH $\uparrow$ | MMLU $\uparrow$ | BBH $\uparrow$ | |
| --- | --- | --- | --- | --- | --- | --- | --- |
| All | 63.8 | 63.3 | 74.1 | 63.8 | 74.1 | 63.8 | - |
| 5% Random | 64.7 | 60.8 | 74.1 | 63.5 | 74.1 | 63.5 | ✗ |
| 5% LESS (Xia et al.) | 65.7 | 62.6 | **74.3** | 66.3 | **74.3** | 65.0 | ✓ |
| 5% TAROT | **66.0** | **65.0** | 74.1 | 66.9 | 74.2 | 65.3 | ✗ |
| TAROT-OTM | 65.7 (0.13%) | 63.6 (0.21%) | **74.3** (0.09%) | **68.9** (0.13%) | **74.3** (0.13%) | **68.7** (0.21%) | ✗ |

2021), Argoverse 2 (Wilson et al., 2021), nuScenes (Caesar et al., 2020), and nuPlan (H. Caesar, 2021). The nuScenes training set ($32k$ samples) serves as the target dataset, while the candidate pool comprises WOMD, Argoverse 2, and nuPlan. From nuPlan, we filter trajectories with a moving distance over 2 meters, yielding $1000k$ samples. We use the official training splits of WOMD and Argoverse 2, including $2000k$ samples. The evaluation is conducted on the nuScenes validation set. This cross-dataset task involves challenges like non-overlapping geographic regions and varying annotation formats. We explore whether TAROT can effectively curate multi-source data to enhance nuScenes validation performance.

**Baselines.** We utilize the same baselines in Section 4.2.

**Data Selection and Model Training.** We employ Auto-Bots (Girgis et al., 2021), a lightweight ($1.5M$ parameters) attention-based prediction model, for data selection. Autobots is pretrained on candidate and target datasets for 50 epochs, and gradients from the final checkpoint guide selection. We evaluate selection ratios of 5%, 20%, 50%, and OTM. OTM selects approximately 31%, 53%, and 22% from WOMD, Argoverse 2, and nuPlan, covering 29.8% of the full dataset. The selected data is used to train the Way-former (Nayakanti et al., 2023) model ($15M$ parameters) for 100 epochs. To highlight that TAROT can also achieve performance gains with reduced training data and iterations, the data weighting method (Section 3.4) is not applied.

**Evaluation Metric.** We use Brier Minimum Final Displacement Error (Brier-minFDE) metric for evaluation, which is used in the Argoverse 2 (Wilson et al., 2021) challenge. We report the best validation performance for each method.

**Main Results.** Figure 7 compares the performance of Way-former trained on datasets with different selection ratios. Key findings include:

- **Effectiveness of TAROT:** TAROT consistently outperforms other baselines. Influence-estimation based heuristics, such as LESS and DsDm, perform well at low selection ratios (5%) but lose effectiveness as the ratio increases, emphasizing the benefits of the distribution-matching ap-

proach. Additionally, OTM selection outperforms training on the entire dataset and other selection ratios, delivering global optimal results. Moreover, TAROT archives the top 1 performance on the nuScenes leaderboard. Results are shown in the Appendix D.

- **Transferability of TAROT:** Data selected by Autobots significantly enhances the performance of Wayformer, a model with $10\times$ the capacity. Furthermore, despite differences in loss functions and model architectures between the two models, the selected data still yields performance improvements. This highlights that although TAROT leverages a pretrained model for selection, the selected dataset retains strong model-agnostic properties.

### 4.4. TAROT for Instruction Tuning

Previous experiments demonstrate that TAROT is particularly effective when dealing with complex target distributions. In this section, we evaluate whether TAROT can also deliver strong performance in LLM tasks, where the target distribution is relatively simple and existing methods have already shown high effectiveness.

**Datasets.** To ensure a fair comparison and maintain consistency with established benchmarks, we replicate the experimental setup described by Xia et al.. We utilize the same candidate dataset, comprising FLAN V2 (Longpre et al., 2023), COT (Wei et al., 2022), DOLLY (Conover et al., 2023) and OPEN ASSISTANT 1 (Köpf et al., 2024), with MMLU (Hendrycks et al., 2021b;a) and BBH (Suzgun et al., 2023) serving as the target tasks for evaluation.

**Data Selection and Model Training.** We conduct experiments using LLAMA-3.1-8B (Dubey et al., 2024) and QWEN-2.5-7B (Team, 2024) as the base models for both data selection and fine-tuning. To evaluate the transferability of our selection method, we also fine-tune a QWEN-2.5-7B using data samples selected by LLAMA-3.1-8B. Data selected by TAROT-OTM is weighted using OT potentials (Section 3.4) with $R = 0.5\%N$. More details are provided in the supplementary materials.

**Evaluation Metric.** We employ Exact Matching (EM) for BBH and Accuracy for MMLU as evaluation metrics.

Table 2: Ablation Study of **WFD** on nuScenes with ensemble size ranging from 1 to 50. Evaluation metric is LDS.

| Ensemble size | 1 | 10 | 50 |
|---|---|---|---|
| w/o whitened | 0.007 | 0.010 | 0.017 |
| w/o norm | 0.024 | 0.060 | 0.097 |
| w/o whitened+norm | 0.005 | 0.008 | 0.015 |
| **WFD** | **0.028** | **0.065** | **0.098** |

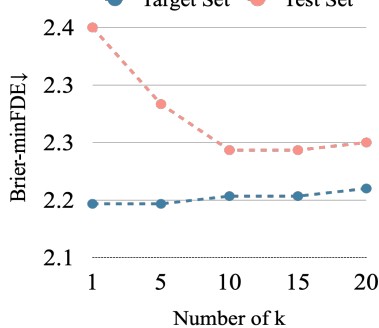

Figure 8: **Ablation study for the choice of k.** The figure shows the performance of Wayformer under different k.

**Main Results.** Table 1 represents the performance of TAROT compared with baselines. Among them, LESS requires the subtask labels of target samples, whereas TAROT operates without this requirement. Our key findings include:

- **TAROT is effective, generalizable, and transferable.** TAROT consistently outperforms all baseline methods and models trained on the full dataset without relying on subtask labels. Besides, data samples selected by LLAMA-3.1-8B enhances performance for QWEN-2.5-7b, indicating strong transferability across different model architectures.
- **Effectiveness of OTM.** Remarkably, TAROT-OTM selects less than 0.5% of the data while achieving performance that is comparable to or even surpasses that of methods selecting 5% of the data. The results suggest that only a small fraction of the candidate dataset is pertinent to the target tasks, and TAROT-OTM successfully identifies these critical samples, thereby significantly reducing training costs without compromising performance.

### 4.5. Computational Complexity

The computational complexity of TAROT mainly arises from two aspects: (1) gradient feature computation and (2) data selection. The former shares the same complexity as previous methods that use projected gradients (Park et al., 2023; Liu et al., 2024; Engstrom et al., 2024). The latter introduces additional computational overhead due to computation of OT distance. We compare the wall-clock runtime of TAROT with baselines and provide a detailed asymptotic complexity analysis in the Appendix G.

### 4.6. Ablation Study

**Whitened Feature Distance.** We conduct an ablation study to analyze the impact of key components on **WFD**. The

results are presented in Table 2. Specifically, w/o whitened indicates that we skip the whitening step and instead use the original projected gradients to compute the distance. w/o norm means that we compute the L2 distance of the gradient vectors without normalization. The results demonstrate that removing these proposed components leads to a performance drop, highlighting the effectiveness of our influence estimation design.

**K-Fold Selection in OTM.** As introduced in Section 3.3, TAROT employs k-fold cross-validation for OTM, where a larger $k$ results in less alignment with the target distribution. We investigate how the choice of $k$ affects data selection in the motion prediction setting (Section 4.3). Specifically, we perform experiments with OTM selection using $k$ values ranging from 1 to 20, evaluating the model's performance on both the test set and the target set. The results, shown in Figure 8, indicate that as $k$ increases, the model's performance on the test set improves, reaching its peak at $k = 10$. Meanwhile, performance on the target set steadily declines, suggesting that the selected data becomes less aligned with the target distribution as $k$ increases.

## 5. Related Work

We review the progress on data selection in this section. More related work is deferred to App. A. Coreset selection aims to identify a dataset subset for optimal in-domain performance (Xia et al., 2022; Mirzasoleiman et al., 2020; Borsos et al., 2020; Killamsetty et al., 2021; Guo et al., 2022; Xiao et al., 2024), assuming validation and candidate data share the same distribution. Our work shifts focus to targeted selection, optimizing performance on a distinct target domain. Prior methods often rely on influence-estimation heuristics: Engstrom et al. (2024) leveraged linear data-model scores (Park et al., 2023) to select data for LLMs pretraining; Xia et al. leveraged influence estimation to select data for LLMs fine tuning. Hu et al. (2024) highlighted these methods' limitations and proposed iterative heuristics. A concurrent work (Liu et al.) also frames data selection as an optimal transport problem. However it still follows the same gradient distance estimation in (Xia et al.) and uses a continuous OT-based formulation, making it less efficient and effective. Our approach introduces a novel distance estimation method and a greedy subset selection pipeline to address these challenges.

## 6. Conclusion

This paper presents TAROT, a general targeted data selection framework leveraging OT theory. By introducing **WFD**, TAROT mitigates biases from dominant features and enhances influence estimation. Through OT-based selection, it aligns subsets closely with target distributions, consistently outperforming traditional methods in tasks such as semantic segmentation, motion prediction, and instruction tuning. These results demonstrate TAROT's effectiveness

in improving domain-specific performance with smaller, more relevant datasets. **Limitations:** In cases with small or highly specific target datasets, TAROT may overfit by overly tailoring the selected data, potentially compromising generalization. A promising future direction is incorporating distribution diversification into the selection objective.

## Acknowledgments

The project was partially funded by Honda R&D Co., Ltd, Valeo Paris, and the Swiss National Science Foundation under the Grant P500PT 222293.

## Impact Statement

This paper presents TAROT, a targeted data selection framework using Optimal Transport to improve model efficiency and performance across diverse tasks. By addressing biases in influence-based heuristics, TAROT enables better dataset alignment with target distributions, reducing computational costs and potential environmental impact. This approach can enhance data efficiency, benefiting scenarios where data privacy or accessibility is a concern. However, like any selection strategy, it may introduce biases if not carefully managed. Future research should explore its implications on fairness and generalization, but overall, TAROT represents a step toward more efficient and responsible data selection in machine learning.

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

# A. More Related Work

**Data Influence Estimation.** Pruthi et al. (2020) proposed a method to estimate data influence by tracing the gradient descent process. Another approach utilizes *datamodels*: Ilyas et al. (2022) showed that simple linear datamodels can predict model outputs effectively. Building on this, Park et al. (2023) introduced a scalable technique for estimating datamodel scores. Subsequent works have optimized these methods, focusing on checkpoint ensembles (Deng et al., 2024), efficient gradient storage (Zhang et al., 2024), and scalable gradient projection (Choe et al., 2024). Our method extends Pruthi et al. (2020) by enhancing estimation robustness through feature covariance whitening.

**Optimal Transport for Data Selection.** Optimal transport has been used for dataset similarity estimation (Alvarez-Melis & Fusi, 2020) and sensitivity analysis in data valuation (Just et al., 2023). Kang et al. (2024) applied it to optimize data selection ratios across multiple sources. We integrate influence estimation with optimal transport, enabling effective subset selection to boost target domain performance.

# B. More Implementation Details

## B.1. Influence Estimation

**ZCA Whitening.** As discussed in Section 4.1, we use ZCA whitening to achieve higher LDS scores due to its ability to preserve the data structure. However, our empirical analysis revealed that ZCA whitening has minimal impact on data selection performance. Therefore, for the data selection experiments, we opted for the more computationally efficient Cholesky whitening.

To apply ZCA whitening, we first compute the covariance matrix:

$$\Sigma = \frac{1}{N} \sum_{i=1}^{N} \tilde{\phi}(z)_i (\tilde{\phi}(z)_i)^\top, \tag{15}$$

where $\tilde{\phi}(z)_i$ represents the centered input features. Next, we perform eigenvalue decomposition:

$$\Sigma = U \Lambda U^\top, \tag{16}$$

where $U$ is the matrix of eigenvectors, and $\Lambda$ is the diagonal matrix of eigenvalues. The ZCA whitening transformation is then defined as:

$$\phi(z)^{\text{ZCA}} = U \Lambda^{-\frac{1}{2}} U^\top \tilde{\phi}(z), \tag{17}$$

which ensures that the transformed features are uncorrelated and have unit variance.

The key difference between ZCA whitening and Cholesky-based whitening lies in the structure of the transformation matrix. In Cholesky whitening, the lower triangular matrix $L$ introduces decorrelation sequentially, while ZCA whitening uses a symmetric transformation to minimize distortion

Table 3: Training Hyperparameters for Semantic Segmentation

| Hyperparameter | Value |
| --- | --- |
| Batch Size | 16 |
| Learning Rate | $1 \times 10^{-4}$ |
| Optimizer | SGD |
| Momentum | 0.9 |
| Weight Decay | $1 \times 10^{-4}$ |
| Learning Rate Scheduler | WarmupPolyLR |
| Warmup Factor | $1/3$ |
| Warmup Method | Linear |

Table 4: Experiment Settings for Motion Prediction

| Data Configurations | |
| --- | --- |
| History Trajectory Time Length | 2.1 seconds |
| Future Trajectory Time Length | 6 seconds |
| Object Type | Vehicle |
| Line Type | Center Lane |
| **Model Configurations** | |
| **Autobots** | |
| Batch Size | 256 |
| Learning Rate | 0.00075 |
| Optimizer | Adam |
| Gradient Clipping | 5 |
| **Wayformer** | |
| Batch Size | 128 |
| Learning Rate | $1 \times 10^{-4}$ |
| Optimizer | AdamW |
| Gradient Clipping | 5 |
| Learning Rate Schedule | [10, 20, 30, 40, 50] |

of the original data structure. This preservation of the data's original geometric structure may result in a better LDS score. However, this structural similarity has a limited impact on data selection tasks, where the primary goal is to identify representative or diverse samples. Since both whitening methods achieve decorrelation and normalization, the relative distances and informativeness of the samples, which are key factors in data selection, remain largely unaffected by the choice of whitening method.

**Linear Data Modeling Score (LDS).** As mentioned in Sec. 4.1, we employ LDS as evaluation metric for influence estimation. LDS is computed by the Spearman rank correlation between the model outputs of models trained on randomly selected subsets and the average of attribution scores as predictions of those model outputs. For Cifar-10, we directly reuse the provided outputs of models trained on different subsets in the Github repository of TRAK (Park et al., 2023), and utilize these outputs to calculate LDS. For

Table 5: Training Hyperparameters for Instruction Tuning

| Hyperparameter | Value |
|---|---|
| Batch Size | 128 |
| Learning Rate | $2 \times 10^{-5}$ |
| Optimizer | Adam |
| LoRA rank | 128 |
| LoRA $\alpha$ | 512 |
| dropout | 0.1 |
| Warmup Epoch | 4 |

nuScences, we randomly select 100 subsets of the candidate datasets. Each subset is sampled to be 50% of the size of the candidate dataset. Then we independently train 100 models on these subsets and utilize their model outputs to calculate LDS.

---

**Algorithm 1** Fixed-Size Selection

---

**Require:** • Candidate dataset $\mathcal{D}_c = \{z_c^{(i)}\}_{i=1}^N$
- Target dataset $\mathcal{D}_t = \{z_t^{(i)}\}_{i=1}^M$
- Desired subset size $S = N$
- Distance metric $d_{\mathcal{Z}}$

**Ensure:** Selected subset $\mathcal{D}_s \subseteq \mathcal{D}_c$ of size $S$
1: Initialize $\mathcal{D}_s \leftarrow \emptyset$
2: Initialize iteration counter $k \leftarrow 1$
3: **while** $|\mathcal{D}_s| < S$ **do**
4:     Identify the $k$-th nearest candidates for each $z_t \in \mathcal{D}_t$ based on $d_{\mathcal{Z}}$
5:     Let $\mathcal{D}_k$ be the set of these candidates
6:     **if** $|\mathcal{D}_s| + |\mathcal{D}_k| \leq S$ **then**
7:         $\mathcal{D}_s \leftarrow \mathcal{D}_s \cup \mathcal{D}_k$
8:     **else**
9:         Compute potential $\phi_i$ for each $z_i \in \mathcal{D}_k$ using:

$$\phi_i = \min_{\pi \in \Pi(\mathcal{D}_s \cup \{z_i\}, \mathcal{D}_t)} \int_{\mathcal{D}_s \cup \{z_i\} \times \mathcal{D}_t} d_{\mathcal{Z}}(z, z_t) \, \mathrm{d}\pi(z, z_t)$$

10:         Rank candidates in $\mathcal{D}_k$ by $\phi_i$ in ascending order
11:         Add $z_i \in \mathcal{D}_k$ to $\mathcal{D}_s$ if it is among the top $S - |\mathcal{D}_s|$ candidates in $\mathcal{D}_k$
12:         **Break** the loop
13:     **end if**
14:     $k \leftarrow k + 1$
15: **end while**
16: Compute potential $\phi_i$ for each $z_i \in \mathcal{D}_s$ to serve as weight $w_i$ and scale to integers.
17: **return** $\mathcal{D}_s, \{w_i\}$

---

### B.2. Semantic Segmentation

Table 3 shows detailed hyperparameters for the semantic segmentation experiments.

---

**Algorithm 2** OT-Distance Minimization Selection (OTM)

---

**Require:** • Candidate dataset $\mathcal{D}_c = \{z_c^{(i)}\}_{i=1}^N$
- Target dataset $\mathcal{D}_t = \{z_t^{(i)}\}_{i=1}^M$
- Number of folds $K$
- Distance metric $d_{\mathrm{OT}}$

**Ensure:** Selected subset $\mathcal{D}_s \subseteq \mathcal{D}_c$
1: Initialize $\mathcal{D}_s \leftarrow \emptyset$
2: Split $\mathcal{D}_t$ into $K$ equal subsets: $\mathcal{D}_t^{(1)}, \mathcal{D}_t^{(2)}, \ldots, \mathcal{D}_t^{(K)}$
3: **for** $fold = 1$ to $K$ **do**
4:     Let $\mathcal{D}_t^{(fold)}$ be the current fold
5:     Let $\mathcal{D}_t^{(\mathrm{tar})} \leftarrow \mathcal{D}_t \setminus \mathcal{D}_t^{(fold)}$
6:     Initialize temporary subset $\mathcal{D}_s^{(fold)} \leftarrow \emptyset$
7:     Initialize iteration counter $k \leftarrow 1$
8:     Initialize Distance $d_{pre} \leftarrow$ 'inf'
9:     **while** True **do**
10:         Let $\mathcal{D}_k$ be the set of $k$-th nearest candidates for each $z_t \in \mathcal{D}_t^{(\mathrm{tar})}$ based on $d_{\mathcal{Z}}$
11:         **if** $d_{\mathrm{OT}}(\mathcal{D}_s^{(fold)} \cup \mathcal{D}_k, \mathcal{D}_t^{(fold)}) > d_{pre}$ **then**
12:             **Break** the loop
13:         **end if**
14:         $\mathcal{D}_s^{(fold)} \leftarrow \mathcal{D}_s^{(fold)} \cup \mathcal{D}_k$
15:         $d_{pre} \leftarrow d_{\mathrm{OT}}(\mathcal{D}_s^{(fold)}, \mathcal{D}_t^{(fold)})$
16:         $k \leftarrow k + 1$
17:     **end while**
18:     $\mathcal{D}_s \leftarrow \mathcal{D}_s \cup \mathcal{D}_s^{(fold)}$
19: **end for**
20: Compute potential $\phi_i$ for each $z_i \in \mathcal{D}_s$ to serve as weight $w_i$ and scale to integers.
21: **return** $\mathcal{D}_s, \{w_i\}$

---

### B.3. Motion Prediction

We use the UniTraj (Feng et al., 2024) framework for motion prediction experiments. Detailed experiment settings are provided in Table 4.

### B.4. Instruction Tuning

We follow the settings and hyperparameters in LESS (Xia et al.). Details are provided in Table 5.

## C. Details for Algorithms

We present the detailed data selection algorithm described in Section 3.3 in Algorithms 1 and 2

## D. NuScenes Leaderboard Performance

In the motion prediction task, TAROT helps Wayformer achieve the first place on the nuScenes leaderboard (Table 7). surpassing MTR-UniTraj (Feng et al., 2024), which relies on large-scale datasets ($2000k$ samples). These results

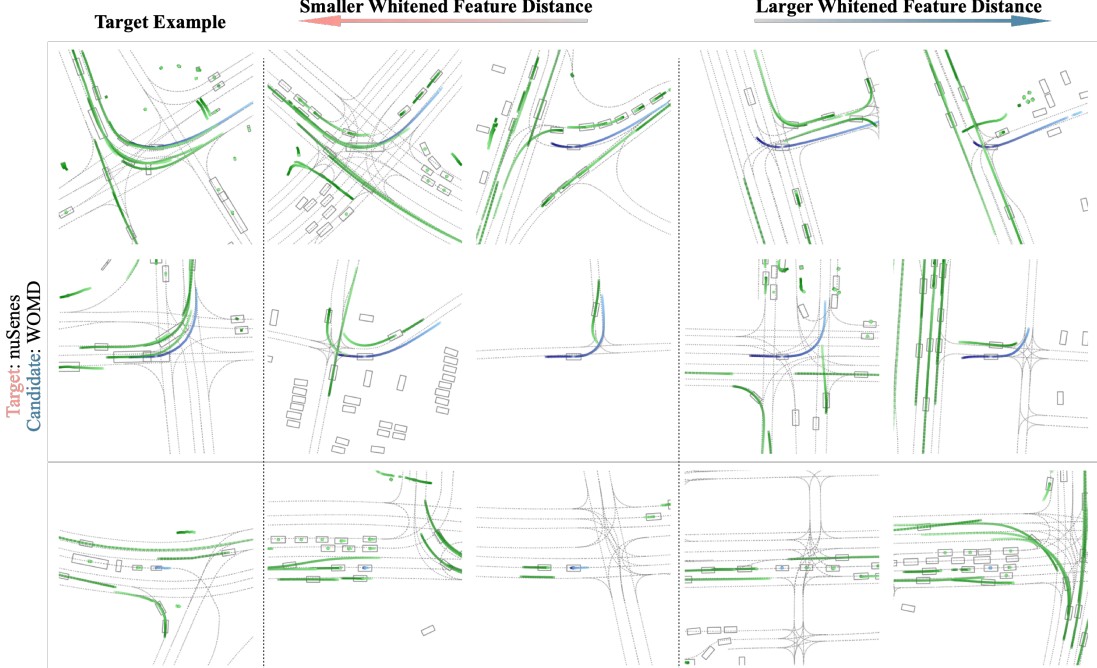

Figure 9: **More Qualitative Results of WFD in Motion Prediction.** Visualization of motion prediction tasks using the nuScenes and Waymo datasets. The dark-to-light blue lines indicate the past (input) and future (ground truth) trajectories for the target vehicle, while green lines represent surrounding vehicle trajectories.

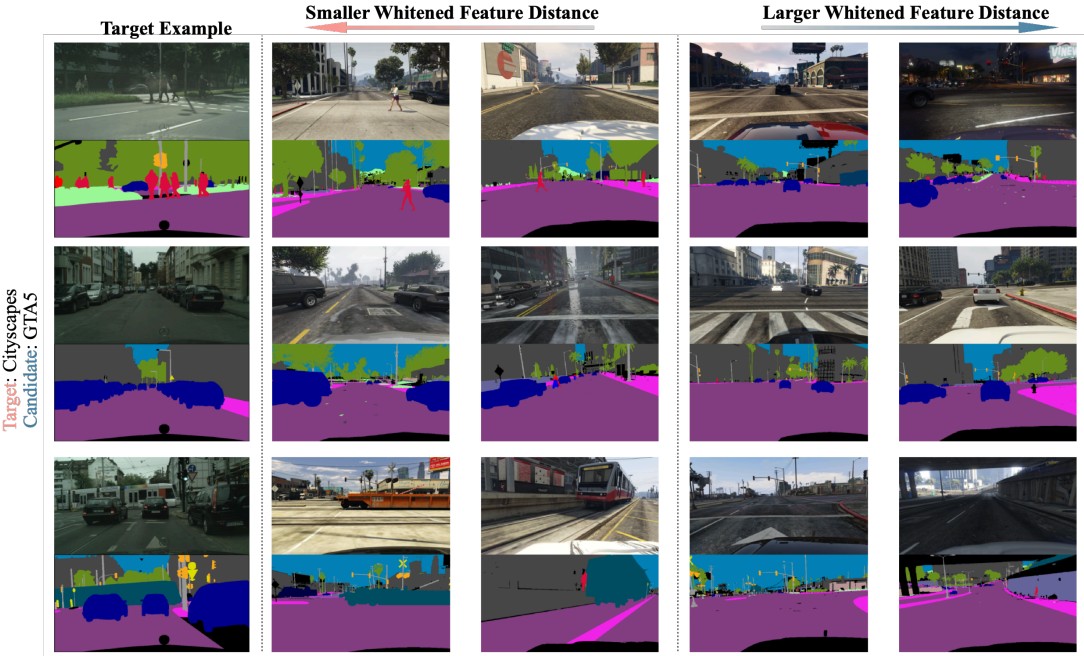

Figure 10: **More Qualitative Results of WFD in Semantic Segmentation.** Semantic segmentation tasks with examples from the Cityscapes and GTA5 datasets. For each target example, the top image shows the RGB scene, and the bottom image depicts the corresponding ground truth semantic segmentation mask.

Table 6: Performance comparison (↑ means higher is better). All methods use 5% of the full training data. TAROT-OTM shows relative improvement over TAROT-Fixed in parentheses.

| Dataset | LLaMA-3.1-8B MMLU ↑ | LLaMA-3.1-8B BBH ↑ | Qwen-2.5-7B MMLU ↑ | Qwen-2.5-7B BBH ↑ |
|---|---|---|---|---|
| 5% LESS | 65.7 | 62.6 | 74.3 | 66.3 |
| 5% TSDS | 65.2 | 63.1 | 73.9 | 66.2 |
| 5% TAROT | **66.0** | **65.0** | 74.1 | 66.9 |
| TAROT-OTM | 65.7 (**0.13%**) | 63.6 (**0.21%**) | **74.3** (**0.09%**) | **68.9** (**0.13%**) |

demonstrate that targeted data selection offers a promising alternative to data scaling for breaking performance boundaries.

# E. Discussion and Comparison with TSDS

TSDS frames data selection as an optimal transport (OT) problem, incorporating a diversity regularizer via kernel density estimation. It selects samples based on distances in gradient embedding space.

In contrast, TAROT uses Whitened Feature Distance (WFD), which removes the confounding effects of gradient covariance and scale. This provides more robust and accurate feature distance estimates, leading to better influence estimation—surpassing the state-of-the-art TRAK

Table 7: **nuScenes Leaderboard.** Evaluated with minADE5 metric: the average of L2 distances between the predicted trajectory and ground truth over the 5 most likely predictions. *Wayformer-All* denotes Wayformer trained with the full candidate dataset (WOMD, Argoverse 2 and nuPlan).

| Method | Ranking (↓) | minADE5 (↓) |
|---|---|---|
| **Wayformer-TAROT** | **1** | **0.92** |
| Wayformer-All (Nayakanti et al., 2023) | 2 | 0.94 |
| MTR-UniTraj (Feng et al., 2024) | 3 | 0.96 |
| Goal-LBP (Yao et al., 2023) | 4 | 1.02 |
| SemanticFormer (Sun et al., 2024) | 5 | 1.14 |

Table 8: The wall clock runtime (measured as single H100 GPU hours) of TAROT compared with LESS and TSDS on instruction tuning task. The gradient feature computation stage is the same for all the methods.

| | Gradient Features Computation | Data Selection |
|---|---|---|
| LESS | | 46 Seconds |
| TAROT-Fixed | 32 Hours | 59 Seconds |
| TAROT-OTM | | 118 Seconds |
| TSDS | | 10 Hours |

Table 9: Asymptotic complexity analysis.

| | Gradient Features Computation | Fixed-Size/OTM Selection |
|---|---|---|
| **Complexity** | $\mathcal{O}(|\mathcal{D}| \cdot N)$ | $\mathcal{O}(|\mathcal{D}| \cdot |\mathcal{D}_{\text{tar}}| \cdot d)$ |

method. Besides, while TSDS uses a continuous OT-based formulation and supports subset size control via sampling, TAROT differs in its greedy, deterministic subset selection that explicitly minimizes the empirical OT distance at each iteration. This enables stronger control over the selected subset and much faster speed for data selection. Indeed, we empirically found that TSDS costs significantly more time than TAROT. The wall-clock time analysis is shown in Table 8. Finally, TAROT introduces an early stopping criterion for OT-based selection via tracking OT distance increase, allowing estimation of optimal selection ratios, which TSDS does not address.

We also include experimental comparisons with TSDS, following the same evaluation settings as described in Section 4.4. The results are shown in Table 6.

# F. More Qualitative Results of WFD

We provide additional qualitative results of **WFD** in Figures 9 and 10. One clear observation is that the most supportive samples tend to closely resemble the target sample in both the input and output spaces. In motion prediction, the most negative samples also share similarities with the target sample in the input space (e.g., map and history trajectories); however, their ground truth trajectories often exhibit minor differences. For instance, in the first row of samples, the most negative examples demonstrate behaviors such as turning left or driving in the right-most lanes, whereas the target sample follows a trajectory in the lane adjacent to the right-most lane. These qualitative findings align with the definition of influence estimation: the most positive samples guide the model to make predictions similar to the target sample, while the negative samples encourage divergent predictions.

# G. Computational Complexity Analysis

Table 9 presents the asymptotic complexity of TAROT. The most computationally expensive step is the computation of gradient features, with its cost scaling linearly with the candidate dataset size $|\mathcal{D}|$, the number of checkpoints $N$, and the gradient dimension $d$. However, the computational costs incurred during these two stages are one-time expenses that can be amortized across multiple target tasks. The data

selection process requires minimal computation.

Table 8 demonstrates the wall-clock runtime (in single H100 GPU hours) of TAROT compared to LESS and TSDS for the instruction tuning task. The gradient feature computation time is identical for both methods. For data selection, both TAROT-Fixed and LESS complete in under one minute, while TAROT-OTM incurs an additional runtime of approximately one minute.

