# OpenReview forum: "TAROT: Targeted Data Selection via Optimal Transport"
_ICML.cc/2025/Conference — ICML 2025 poster_

### Official Review · Reviewer_oARp · 2025-03-10

**Overall Recommendation:** 3

**Summary:**

This paper proposes to address the problem of task-specific sample selection from the perspective of distribution matching that can be solved by optimal trasnport. Experiments on influence estimation, semantic segmentation, motion prediction, and instruction tuning show the effectivenss of the proposed method.

**Claims And Evidence:**

Please refer to the weakness.

**Essential References Not Discussed:**

The key contribution is an OT-driven task-specific sample selection paradigm. However, a closely related paper has been overlooked [1], which also employs OT for task-specific data selection and can be based on either semantic embedding or gradient (in particular, this submission paper adpots gradient). On the other hand, both this submission and [1] utilize nearest neighbor search during OT-based sample selection in order to reduce computational cost. in my oponion, it's necessary to discuss the connections and differences between this submission and [1], as well as to provide experimental comparisons with [1].

[1] TSDS: Data Selection for Task-Specific Model Finetuning. NIPS 2024.

**Experimental Designs Or Analyses:**

The author did not report the results of TYDIQA like LESS and [1] did. Could the author please explain the reason for this? Is it due to experimental limitations or other considerations? It is necessary to clarify this to make the research more complete and comparable with existing works.

[1] TSDS: Data Selection for Task-Specific Model Finetuning. NIPS 2024.

**Methods And Evaluation Criteria:**

Yes. But lack of some important baselines. Please refer to the weakness and experimental design.

**Other Comments Or Suggestions:**

Please refer to the Weakness.

**Other Strengths And Weaknesses:**

Strengths:

1. The introduction of OT to task-specific data selection is reasonable.
2. Extensive experiments on different settings prove the effectiveness.


Weakness:

1. The author attempts to introduce more content into the paper, which makes it difficult to grasp the key points of the paper. For example, the author mentions a variant of the fixed-size selection method (TAROT-FSS), and the experimental results corresponding to it are TAROT-5%/20%/50%? Also, regarding data weighting (Sec.3.4), is it used to weight the samples when constructing the marginal distributions alpha or beta? The definition and significance of this part are not very clear. Moreover, the repetition factor is only used in Section 4.2. Why is it not considered in Sections 4.3 and 4.4? The experimental setup is somewhat confusing.

2. The improvement in experimental results compared to LESS seems minimal, especially in Figures 5 and 7. However, Table 7 shows that TAROT requires significantly more computational cost and time than LESS. This makes the method in the paper appear to lack a good trade-off between performance improvement and computational cost.

3. The author mentions in the abstract (Line 016) that "These methods perform well on limited, unimodal data (i.e., data following a single pattern) but become less effective as target data increases in complexity." However, I don't see design or experiment on sample selection for multi-modal datasets in this paper. Perhaps I missed some details. Maybe Motion Prediction? The Motion Prediction task (Waymo) involves multiple modalities, such as RGB and point clouds. However, from a task setting perspective, motion prediction based on historical data is often considered a single-modality task. Also, the experimental details seem to lack clarification on which modalities were used and how different input data types interacted during the data selection.

4. When introducing OT, the author uses the integral form under continuous distributions in Eq. 3. However, when formulating the marginal distributions alpha and beta, a discrete form is used. Considering that the author employs entropic OT (Sinkhorn), it would be more consistent to use the discrete form of OT notation throughout.

5. Line 161: "due to the correlation of gradient feature" lacks necessary references, experiments, or observations to support it. This makes the motivation for the Whitened Feature Distance not entirely convincing.

6. Cholesky whitening lacks relevant references.

**Questions For Authors:**

What's the implementation of $c(z, z_t)$ in Eq.13? Cosine distance or WFD $d_{\mathcal{Z}}^{w}(z,z')$?

Please refer to the weakness.

**Relation To Broader Scientific Literature:**

Please refer to the weakness and essential references.

**Theoretical Claims:**

Yes. correct.

---

> ### Author Rebuttal · Authors · 2025-03-30
>
> > ### Q3: Clarifying Modality
>
> By **modality**, we refer to **distributional modality**, rather than input modality. We elaborate on why other tasks exhibit greater distributional multi-modality compared to **instruction tuning**, as summarized in the table below:
>
> |               | Motion Prediction                               | Semantic Segmentation                                        | Instruction Tuning           |
> |---------------------|--------------------------------------------------|---------------------------------------------------------------|------------------------------|
> | Target Dataset Size | 32,186                                           | 2,975                                                         | 81 (BBH) / 285 (MMLU)        |
> | Domain Coverage     | Diverse traffic scenarios from two cities | Stereo video scenes from 50 different cities                  | Specific LLM task types      |
>
> As shown, both motion prediction and semantic segmentation involve significantly larger datasets and broader domain coverage, making data selection inherently more challenging. Our method, validated across multiple tasks, demonstrates better generalization in such multi-modal distributional settings.
>
> ---
>
> > ### Q2: Comparison with LESS
>
> We respectfully disagree with the assessment that TAROT offers only limited improvement over LESS. As discussed above, TAROT yields **substantial gains** on tasks characterized by high distributional multimodality:
>
> - Compared to LESS, **TAROT achieves:**
>   - **+44%** improvement in **semantic segmentation**
>   - **+102%** improvement in **motion prediction**
>   - Even in **instruction tuning**, which has relatively lower modality, TAROT outperforms LESS using just **2%** of the data.
>
> **Computational Cost:**
> While TAROT incurs additional cost due to OT distance computation, this cost is minimal (118 seconds) compared to gradient calculation time (32 hours). Moreover, gradient calculation is a one-time cost and can be amortized across multiple tasks.
>
>
>
> ---
>
> > ### Comparison with TSDS
>
> As noted in our response to Reviewer 5vQX, we discuss differences with TSDS in detail. Here, we provide additional experiments:
>
> | Dataset         | Llama-3.1-8B MMLU ↑ | Llama-3.1-8B BBH ↑ | Qwen-2.5-7B MMLU ↑ | Qwen-2.5-7B BBH ↑ |
> |----------------|---------------------|---------------------|---------------------|---------------------|
> | 5% LESS        | 65.7                | 62.6                | **74.3**            | 66.3                |
> | 5% TSDS        | 65.2                | 63.1                | 73.9                | 66.2                |
> | 5% TAROT       | **66.0**            | **65.0**            | 74.1                | 66.9                |
> | TAROT-OTM      | 65.7 (0.13%)        | 63.6 (0.21%)        | **74.3** (0.09%)    | **68.9** (0.13%)    |
>
> **Time Complexity:**
> We measured the data selection time on a node with 370 GB RAM, 64 CPUs, and an H100 GPU:
>
> | Method             | LESS | TSDS       | TAROT-Fixed | TAROT-OTM  |
> |--------------------|------|------------|-------------|------------|
> | Data Selection Time| 46s  | **10 hrs** | 59s         | 118s       |
>
> **OOM Issue on Motion Prediction:**
> TSDS cannot be applied to the motion prediction task (32k samples) due to out-of-memory (OOM) errors.
>
> **Additional Results on TydiQA:**
> Due to resource constraints, we focused on BBH/MMLU. For completeness, we now include results on TydiQA (only 9 target samples):
>
> | Dataset                | All | 5% Random | 5% LESS | 5% TAROT | TAROT-OTM         |
> |------------------------|-----|-----------|---------|----------|-------------------|
> | Llama-3.1-8B TydiQA ↑  | 63.1| 61.0      | 69.2    | **71.1** | 66.9 (0.05%)      |
>
> ---
> > ### Q5: Motivation for Whitened Feature Distance
>
> We visualized the **[covariance matrix of raw gradient features](https://postimg.cc/SXSkcZrX)**, which reveals strong correlations—supporting our motivation for feature whitening. Additionally, this is conceptually aligned with the findings in _“Whitening for Self-Supervised Representation Learning”_, which we will cite in the revision.
>
> ---
> > ### Q1: Clarifications
>
> We appreciate the feedback and will revise the manuscript for clarity. Here are key clarifications:
>
> - **Fixed-Size vs. OTM:**
>   The 5%, 10%, 20%, and 50% results refer to **TAROT-Fixed**. **TAROT-OTM** dynamically selects the ratio that minimizes OT distance.
>
> - **Data Weighting Implementation:**
>   Weighting is applied by **repeating samples** during training based on their assigned weights.
>
> - **Training Overhead:**
>   Due to increased computation, we omit data weighting in the **motion prediction** task to demonstrate TAROT’s effectiveness without any training overhead.
>
> - **Equation 13:**
>   Refers to the **Whitened Feature Distance (WFD)**.
>
> ---
>
> > ### Q4 & Q6
>
> Thank you for the suggestions. We will ensure consistent use of the discrete formulation throughout the paper and include citations in the revised version.

---

> > ### Comment · Reviewer_oARp · 2025-04-03
> >
> > Thanks for the authors feedback. I don't have other concerns and will increase my rating accordingly.

---

### Official Review · Reviewer_UG2a · 2025-03-13

**Overall Recommendation:** 4

**Summary:**

The paper introduces a framework for targeted data selection by minimizing the distance between selected data and target data distribution. The method addresses the limitations of existing influence-based greedy heuristics, which does not perform well on multimodal data distributions. The framework is evaluated across multiple tasks, including semantic segmentation, motion prediction, and instruction tuning, demonstrating consistent improvements over state-of-the-art methods. The authors also provide a detailed analysis of the computational complexity and ablation studies to validate the effectiveness of their approach.

## Update after rebuttal
Thank you for the responses. I keep my original score of 4.

**Claims And Evidence:**

Yes, the claims are supported by clear evidence.

**Essential References Not Discussed:**

I am not aware of significant missing references that are essential to understanding the key contributions of the paper.

**Experimental Designs Or Analyses:**

While I am not an expert in assessing the validity of experimental design within this specific field, the experimental designs appear to adhere to standard practices.

**Methods And Evaluation Criteria:**

I am not deeply familiar with the empirical work and existing evaluation approaches in this field, the evaluation criteria used in the paper either align with established practices from prior research or appear to be well-justified and reasonable.

**Other Comments Or Suggestions:**

The paper is well written.

**Other Strengths And Weaknesses:**

Strengths:
- The paper proposes a novel approach for mitigating a major weakness of the previous data selection method, namely, the summed influence fail to find diverse data. The key technique is novel: they use whitening to decorrelate the features and select data by OT distance minimization. They show by extensive experiments that TAROT mitigates biases from dominant features and outperforms previous methods in tasks including semantic segmentation, motion prediction, and instruction tuning.

Weaknesses:
- The paper follows a line of work in data selection, but does not seem to discuss other existing approaches for similar data selection problems. For example, the literature on data distillation and the following paper,
"Data Valuation using Reinforcement Learning", Jinsung Yoon, Sercan Arik, Tomas Pfister Proceedings of the 37th International Conference on Machine Learning, PMLR.

**Questions For Authors:**

None.

**Relation To Broader Scientific Literature:**

The method addresses the limitations of existing influence-based greedy heuristics (Xia et al.; Engstrom et al., 2024) for data selection, which does not perform well on multimodal data distributions as noted by  (Hu et al., 2024). It is based on the Optimal Transport theory, which was also used in (Just et al., 2023).

**Theoretical Claims:**

The work is mostly empirical and proposes no major theoretical claims.

---

> ### Author Rebuttal · Authors · 2025-03-30
>
> **Difference Between Data Distillation and Valuation**
>
>
>
> Thank you for this insightful comment. We appreciate the opportunity to clarify the relationship between TAROT and other data selection paradigms, particularly **Data Valuation via Reinforcement Learning (DVRL)** [Yoon et al., ICML 2020] and **Data Distillation**.
>
>
> **1. Comparison to DVRL (Data Valuation via Reinforcement Learning)**
>
>
>
> DVRL frames data valuation as a meta-learning problem, where a Data Value Estimator (DVE) is trained using reinforcement learning (RL) to assign importance weights to training samples. The reward signal is derived from the performance of a predictor model on a validation set. This setup allows DVRL to dynamically prioritize “useful” samples and has shown promising results in settings with noisy labels or domain shifts.
>
>
>
> **TAROT departs from DVRL in several key aspects:**
>
> • **Targeted vs. General Data Value Estimation:**
>
> DVRL learns instance-wise importance scores primarily to boost general predictive performance. In contrast, TAROT is explicitly designed for _targeted_ selection: it identifies a subset of data that minimizes the **optimal transport (OT) distance** to a distinct target distribution. This enables TAROT to adapt to complex, potentially multimodal target domains—something DVRL is not explicitly formulated to handle.
>
> • **Model-Agnostic Transferability:**
>
> TAROT computes data selection using WFD-based embeddings from a lightweight model, but supports downstream training with larger, task-specific models. As demonstrated in Section 4.3, the selected data generalizes across diverse architectures and tasks (e.g., motion prediction, instruction tuning). DVRL, by contrast, couples valuation tightly to the predictor, reducing flexibility for out-of-distribution generalization or transfer across models.
>
> • **Computational Simplicity and Scalability:**
>
> DVRL’s RL-based training is often computationally intensive and sensitive to hyperparameters, especially in high-dimensional settings. TAROT employs a more efficient and stable OT-based greedy selection method, using gradient feature whitening and normalization to reduce dominant component bias. As shown in our runtime analysis (Appendix F), TAROT scales effectively to large datasets (e.g., 2M motion samples) without requiring complex policy optimization.
>
> ----------
>
> **2. Comparison to Data Distillation Approaches**
>
>
>
> Data distillation typically involves synthesizing or filtering datasets to mimic the performance of a teacher model. Although both data distillation and TAROT aim to curate more efficient training sets, their approaches and goals differ significantly:
>
> • **Objective Focus:**
>
> •  _Data Distillation_ is model-centric, often relying on teacher-student frameworks or aligning predictions/logits.
>
> •  _TAROT_ is distribution-centric, selecting data that explicitly aligns with the **target distribution** using whitened OT distances computed over gradient features.
>
> • **Dependency on a Teacher Model:**
>
> TAROT does not require a high-performing teacher model. Instead, it operates in scenarios where such a model may not exist, making it broadly applicable.
>
> • **Automatic Selection Ratio:**
>
> TAROT automatically infers optimal selection ratios through OT distance minimization (Section 3.3), a capability typically absent in distillation-based methods.
>
> ----------
>
> Thank you again for this constructive suggestion. It has helped us better articulate TAROT’s position within the broader landscape of data selection research.

---

### Official Review · Reviewer_5vQX · 2025-03-13

**Overall Recommendation:** 3

**Summary:**

The paper proposes a data selection method for a specific target domain from a candidate set by posing it as a distribution matching problem. The paper proposes to use whitened gradient features as the base distance to compute the optimal transport between the two sets.  Effectiveness of this proposed method is shown on motion prediction, semantic segmentation and instruction tuning tasks.

## Update after the rebuttal
I thank the authors for addressing my concerns during the rebuttal. I keep my initial rating of 3.

**Claims And Evidence:**

Well supported.

**Essential References Not Discussed:**

1. TSDS: Data Selection for Task-Specific Model Finetuning, NeurIPS 2024
2. Data Selection for Language Models via Importance Resampling, NeurIPS 2023

**Experimental Designs Or Analyses:**

The experimental design is standard and makes sense.

**Methods And Evaluation Criteria:**

Yes.

**Other Comments Or Suggestions:**

NA

**Other Strengths And Weaknesses:**

Strengths
1. The idea of selecting the data from a candidate set to improve performance on a target set via OT is important and relevant for many applications.
2. The idea of using gradient-based features in OT formulation and the use of whitening and normalization seems to be helpful in improving performance on considered baselines.
3. Empirical improvement across a diverse set of application shows the effectiveness of the method

Weaknesses
1. The proposed method is compares only with LESS in the instruction tuning experiment where as comparison with two recent methods TSDS and DSIR should also be included. (The titles of the works are mentioned above.)

2. The amount of target data required to estimate the OT distances is not ablated for the three applications.

3. How many samples from source and target are used to compute the OT distance?

**Questions For Authors:**

See Weaknesses above

**Relation To Broader Scientific Literature:**

The use of OT for dataset selection is very relevant and related to the previous works in the literature.

**Theoretical Claims:**

No theoretical claims made.

---

> ### Author Rebuttal · Authors · 2025-03-30
>
> **Q1. Discussion and Comparison with TSDS and DSIR**
>
> Thank you for the suggestion. We have revised the related work section to better highlight how **TAROT** differs from related methods, particularly **TSDS** and **DSIR**, which also address data selection from a distribution-matching perspective.
>
> **TSDS** frames data selection as an **optimal transport (OT)** problem, incorporating a diversity regularizer via kernel density estimation. It selects samples based on distances in gradient embedding space.
>
> In contrast, **TAROT**:
>
> - Uses **Whitened Feature Distance (WFD)**, which removes the confounding effects of gradient covariance and scale. This provides more robust and accurate feature distance estimates, leading to better **influence estimation**—surpassing the state-of-the-art TRAK method.
> - While TSDS uses a continuous OT-based formulation and supports subset size control via sampling, TAROT differs in its **greedy, deterministic subset selection** that explicitly minimizes the **empirical OT distance** at each iteration. This enables stronger control over the selected subset and much faster speed for data selection. Indeed, we empirically found that TSDS cost significantly more time than TAROT. Please see our results below.
> - Introduces an **early stopping criterion** for OT-based selection via tracking OT distance increase, allowing estimation of **optimal selection ratios**, which TSDS does not address.
>
>
> **DSIR** similarly uses distribution matching but estimates **importance weights in low-dimensional n-gram space**. While efficient, this space lacks the capacity to capture high-level, task-specific semantics.
>
> **TAROT**, on the other hand:
>
> - Operates in **task- and model-specific gradient space**, and performs selection to **explicitly minimize OT distance**, making it more effective for complex or multimodal tasks like motion prediction.
>
> **Summary**: TAROT improves over TSDS and DSIR by (1) more accurate influence estimation via WFD, (2) optimal selection ratio estimation, and (3) stronger performance across domains where simpler feature spaces fall short.
>
> We include **experimental comparisons with TSDS**. Due to time constraints and DSIR’s evaluation overlap with LESS, we do not re-run DSIR experiments.
>
> we provide additional experiments:
>
> | Dataset         | Llama-3.1-8B MMLU ↑ | Llama-3.1-8B BBH ↑ | Qwen-2.5-7B MMLU ↑ | Qwen-2.5-7B BBH ↑ |
> |----------------|---------------------|---------------------|---------------------|---------------------|
> | 5% LESS        | 65.7                | 62.6                | **74.3**            | 66.3                |
> | 5% TSDS        | 65.2                | 63.1                | 73.9                | 66.2                |
> | 5% TAROT       | **66.0**            | **65.0**            | 74.1                | 66.9                |
> | TAROT-OTM      | 65.7 (0.13%)        | 63.6 (0.21%)        | **74.3** (0.09%)    | **68.9** (0.13%)    |
>
> **Time Complexity:**
> We measured the data selection time on a node with 370 GB RAM, 64 CPUs, and an H100 GPU:
>
> | Method             | LESS | TSDS       | TAROT-Fixed | TAROT-OTM  |
> |--------------------|------|------------|-------------|------------|
> | Data Selection Time| 46s  | **10 hrs** | 59s         | 118s       |
>
> **OOM Issue on Motion Prediction:**
> TSDS could not be applied to the motion prediction task (32k samples) due to out-of-memory (OOM) errors.
>
>
> ---
>
> **Q2. Target Data Ablation for OT Distance**
>
> We performed an ablation using the **nuPlan** motion prediction dataset. Candidate data consists of 92k samples from four cities; the target set and test set are 4k and 1k held-out Boston samples respectively. We fix the selection percentage at 10% and vary the amount of target data used from 1k to 4k.
>
> | Target Data / Selected | DsDm | LESS | Random | TAROT |
> |------------------------|------|------|--------|--------|
> | 1000 / 9,200           | 3.12 | 3.07 | 2.77   | 2.33   |
> | 2000 / 9,200           | 2.97 | 3.10 | 2.77   | 2.32   |
> | 3000 / 9,200           | 3.18 | 3.04 | 2.77   | 2.17   |
> | 4000 / 9,200           | 3.00 | 3.08 | 2.77   | 2.14   |
>
> **TAROT’s performance improves steadily** as more target data is used, while baselines show no consistent gains over random selection.
>
> ---
>
> **Q3. OT Distance Sample Counts**
>
> We detail the number of samples used for OT distance computation:
>
> | **Task**                  | **Target Samples** | **Candidate Samples** | **Target/Candidate %** |
> |---------------------------|--------------------|------------------------|-------------------------|
> | Instruction Tuning — MMLU | 285                  | 270,679                 | 0.1%                  |
> | Instruction Tuning — BBH  | 81                 | 270,679                  | 0.03%                   |
>
> These details will be included in the updated manuscript.

---

> > ### Comment · Reviewer_5vQX · 2025-04-02
> >
> > I thank the authors for their response to my questions. I have some follow up questions.
> >
> > 1. What task was used to report the time complexity results. Which step in TSDS is the bottleneck in terms of time? Is the 10hrs number for TSDS just the time for data selection or does it include other steps as well?
> >
> > 2. For the additional experiments, how does the performance of the three methods change when you have lesser data (like 1% or 0.5%). Is TAROT still better?

---

> > > ### Author Response · Authors · 2025-04-08
> > >
> > > Thanks for your questions. We conducted additional experiments to address them.
> > >
> > > **Q1. Time Complexity of TSDS**
> > >
> > > We report the wall-clock time on the instruction tuning task (MMLU dataset, which is more time-consuming than BBH.) The reported time **only includes the data selection step**; all three methods incur the same cost (~32 hours) for gradient computation and caching.
> > >
> > > Upon inspecting the TSDS codebase, we identified **kernel density estimation (KDE)** as the primary bottleneck. KDE requires an extra round of neighbor searches over a large set of samples, resulting in near-quadratic time complexity with respect to the number of candidates. In contrast, TAROT avoids this step entirely through a deterministic OT-based greedy selection procedure, offering significant speed advantages.
> > >
> > > **Q2. Additional Comparison with LESS at Low Selection Ratios**
> > >
> > > Due to time constraints, we ran experiments using the fastest configuration—Qwen-2.5B on the BBH benchmark. We compared TAROT and LESS at finer-grained selection ratios, including 0.13% (OTM ratio), 0.5%, and 1%. Results are shown below and available [here](https://postimg.cc/K1W4q1yx).
> > >
> > > **Qwen-2.5B BBH**
> > >
> > > | Method | 0.13% | 0.5% | 1%   | 5%   |
> > > |--------|-------|------|------|------|
> > > | LESS   | 67.2  | 67.8 | 67.6 | 66.3 |
> > > | TAROT  | 68.9  | 68.5 | 67.5 | 66.9 |
> > >
> > > TAROT consistently outperforms or matches LESS, especially at the OTM ratio. This trend reflects the nature of our OT-based selection: as the selection ratio increases, the chosen subset may gradually drift from the target distribution, leading to diminishing returns. This supports our claim that OT distance serves as a reliable signal for estimating the optimal selection ratio.

---

### Official Review · Reviewer_fLAa · 2025-03-17

**Overall Recommendation:** 4

**Summary:**

The authors formulate targeted data selection as a distribution matching problem and propose a new framework to efficiently select the most suitable training datasets. Massive experiments were conducted to support the effectiveness of the proposed method.

**Claims And Evidence:**

The authors conducted massive experiments and ablation studies to support the effectiveness of each component of the proposed method. However, I found (original: do not find) no supporting evidence (original: clues) for the claim, 'This work identifies two primary factors contributing to this limitation: (ii) the restrictive linear additive assumptions inherent in greedy selection strategies.

**Essential References Not Discussed:**

No

**Experimental Designs Or Analyses:**

I have reviewed the code in the supplementary material. There are no scripts or detailed instructions for reproduction, and I hope the authors can provide the detailed code for reproduction during the review process.

**Methods And Evaluation Criteria:**

Yes, the proposed method is well-aligned with the data selection problem. However, I think that the result about wall-clock time (corrected from wall-lock) should be included to furthur evaluate the proposed method's effectiveness.

**Other Comments Or Suggestions:**

The notation is abused, and I suggest that the author unify the notation system.

**Other Strengths And Weaknesses:**

No

**Questions For Authors:**

No

**Relation To Broader Scientific Literature:**

No

**Theoretical Claims:**

There is no theoretical claims.

---

> ### Author Rebuttal · Authors · 2025-03-30
>
> **Q1: No supporting evidence for this claim: the linear additive assumptions inherent in greedy selection strategies are restrictive.**
>
>
> Thank you for highlighting this. The limitations of linear additive assumptions in influence estimation have been thoroughly examined in the paper _Most Influential Subset Selection: Challenges, Promises, and Beyond_. We summarize their key findings that support our claim:
>
>
>
> **Failure Mode 1: Inaccurate Influence Estimates**
>
>
>
> Even under linear regression, influence estimates can be misleading if the **leverage score** $h_{ii}$ is ignored. The actual individual effect is: $A_{-{i}} = \frac{x_{\text{test}}^\top N^{-1} x_i r_i}{1 - h_{ii}},$
> which differs from the estimate $v_i$ by a factor of $\frac{1}{1 - h_{ii}}$, often resulting in underestimation for high-leverage samples.
>
>
> **Failure Mode 2: Violation of Additivity**
>
>
>
> Even if individual effects are accurately estimated, heuristics such as LAGS (Leverage-Adjusted Greedy Selection) may still fail due to non-additive group influence.
>
>
>
> **Amplification**
>
> Group influence can be _super-additive_ when samples are similar (e.g., duplicates). For $c$ identical samples $(x_i, y_i): \frac{A_{-{i}^c}}{A_{-{i}}} = \frac{c(1 - h_{ii})}{1 - c h_{ii}} > c.$
>
> **Cancellation**
>
> Conversely, group influence can be _sub-additive_, meaning $A_{-{i,j}} < A_{-{j}}$, due to cross-leverage $h_{ij}$ and residual interactions:
>
>
> $$A_{-{i,j}} = \frac{(1 - h_{ii})(1 - h_{jj})(A_{-{i}} + A_{-{j}}) + h_{ij} x_{\text{test}}^\top N^{-1}(x_i r_j + x_j r_i)}{(1 - h_{ii})(1 - h_{jj}) - h_{ij}^2}$$
>
>
> These findings collectively illustrate the core limitations underpinning our claim. Additionally, our empirical comparisons with data selection methods like LESS and DSDM—which rely on linear additive assumptions—reinforce this point. For instance, Figure 6 shows that samples selected by DSDM are concentrated near the center of the target distribution, failing to capture diversity effectively.
>
> ----------
>
> **Q2. Results about Wall-clock time.**
>
>
> We provided wall-clock time results in Table 7 of the appendix. It show sthat TAROT requires comparable time for gradient computation and slightly longer time for data selection compared to baseline methods (1–2 minutes). We attached the table here for your convenience.
>
> |                          | **Gradient Features Computation** | **Data Selection** |
> |--------------------------|-----------------------------------|--------------------|
> | LESS                     | 32 Hours                          | 46 Seconds         |
> |TSDS                     | 32 Hours                          | 65 Minutes          |
> | *TAROT*-Fixed    |     32 Hours                                 | 59 Seconds         |
> | *TAROT*-OTM      |      32 Hours                                | 118 Seconds        |
>
> ----------
>
> **Q3. Providing Detailed Code Instructions**
>
>
>
> Thank you for the suggestion. While we would like to update the code during the review phase, current guidelines do not permit this. We commit to releasing the full code along with detailed instructions upon paper acceptance.
>
> **Notation System**
>
> Thanks for your suggestion. We agree that the current notion system need futher unification and will pay more attention to optimize it.

---

### Decision · Program_Chairs · 2025-05-01

**Decision:**

Accept (poster)

**Comment:**

The authors formulate data selection for training as a distribution matching problem and apply OT to solve it. They conduct extensive experimental evaluations to demonstrate the efficacy of their method. Additionally, they have addressed the main concerns raised by the reviewers, namely wall-clock time and comparison to TSDS. I recommend acceptance.